# Online combinatorial optimization with stochastic decision sets and adversarial losses

**Gergely Neu**      **Michal Valko**
SequeL team, INRIA Lille – Nord Europe, France
{gergely.neu,michal.valko}@inria.fr

## Abstract

Most work on sequential learning assumes a fixed set of actions that are available all the time. However, in practice, actions can consist of picking subsets of readings from sensors that may break from time to time, road segments that can be blocked or goods that are out of stock. In this paper we study learning algorithms that are able to deal with *stochastic availability* of such unreliable composite actions. We propose and analyze algorithms based on the Follow-The-Perturbed-Leader prediction method for several learning settings differing in the feedback provided to the learner. Our algorithms rely on a novel loss estimation technique that we call *Counting Asleep Times*. We deliver regret bounds for our algorithms for the previously studied *full information* and *(semi-)bandit* settings, as well as a natural middle point between the two that we call the *restricted information* setting. A special consequence of our results is a significant improvement of the best known performance guarantees achieved by an efficient algorithm for the sleeping bandit problem with stochastic availability. Finally, we evaluate our algorithms empirically and show their improvement over the known approaches.

## 1  Introduction

In online learning problems [4] we aim to sequentially select actions from a given set in order to optimize some performance measure. However, in many sequential learning problems we have to deal with situations when some of the actions are not available to be taken. A simple and well-studied problem where such situations arise is that of sequential routing [8], where we have to select every day an itinerary for commuting from home to work so as to minimize the total time spent driving (or even worse, stuck in a traffic jam). In this scenario, some road segments may be blocked for maintenance, forcing us to work with the rest of the road network. This problem is isomorphic to packet routing in ad-hoc computer networks where some links might not be always available because of a faulty transmitter or a depleted battery. Another important class of sequential decision-making problems where the decision space might change over time is recommender systems [11]. Here, some items may be out of stock or some service may not be applicable at some time (e.g., a movie not shown that day, bandwidth issues in video streaming services). In these cases, the advertiser may refrain from recommending unavailable items. Other reasons include a distributor being overloaded with commands or facing shipment problems.

Learning problems with such partial-availability restrictions have been previously studied in the framework of prediction with expert advice. Freund et al. [7] considered the problem of online prediction with *specialist experts*, where some experts' predictions might not be available from time to time, and the goal of the learner is to minimize regret against the best *mixture* of experts. Kleinberg et al. [15] proposed a stronger notion of regret measured against the best *ranking* of experts and gave efficient algorithms that work under stochastic assumptions on the losses, referring to this setting as prediction with *sleeping experts*. They have also introduced the notion of *sleeping bandit* problems where the learner only gets partial feedback about its decisions. They gave an inefficient algorithm

for the non-stochastic case, with some hints that it might be difficult to learn efficiently in this general setting. This was later reaffirmed by Kanade and Steinke [14], who reduce the problem of PAC learning of DNF formulas to a non-stochastic sleeping experts problem, proving the hardness of learning in this setup. Despite these negative results, Kanade et al. [13] have shown that there is still hope to obtain efficient algorithms in adversarial environments, if one introduces a certain *stochastic a assumption on the decision set.*

In this paper, we extend the work of Kanade et al. [13] to combinatorial settings where the action set of the learner is possibly huge, but has a compact representation. We also assume *stochastic action availability*: in each decision period, the decision space is drawn from a *fixed but unknown* probability distribution independently of the history of interaction between the learner and the environment. The goal of the learner is to minimize the sum of losses associated with its decisions. As usual in online settings, we measure the performance of the learning algorithm by its regret defined as the gap between the total loss of the best fixed decision-making policy from a pool of policies and the total loss of the learner. The choice of this pool, however, is a rather delicate question in our problem: the usual choice of measuring regret against the best fixed action is meaningless, since not all actions are available in all time steps. Following Kanade et al. [13] (see also [15]), we consider the policy space composed of all mappings from decision sets to actions within the respective sets.

We study the above online combinatorial optimization setting under three feedback assumptions. Besides the full-information and bandit settings considered by Kanade et al. [13], we also consider a restricted feedback scheme as a natural middle ground between the two by assuming that the learner gets to know the losses associated only with *available* actions. This extension (also studied by [15]) is crucially important in practice, since in most cases it is unrealistic to expect that an unavailable expert would report its loss. Finally, we also consider a generalization of bandit feedback to the combinatorial case known as *semi-bandit* feedback.

Our main contributions in this paper are two algorithms called SLEEPINGCAT and SLEEPINGCAT-BANDIT that work in the restricted and semi-bandit information schemes, respectively. The best known competitor of our algorithms is the BSFPL algorithm of Kanade et al. [13] that works in two phases. First, an initial phase is dedicated to the estimation of the distribution of the available actions. Then, in the main phase, BSFPL randomly alternates between exploration and exploitation. Our technique improves over the FPL-based method of Kanade et al. [13] by removing the costly exploration phase dedicated to estimate the availability probabilities, and also the explicit exploration steps in their main phase. This is achieved by a cheap alternative loss estimation procedure called Counting Asleep Times (or CAT) that does not require estimating the distribution of the action sets. This technique improves the regret bound of [13] after $T$ steps from $\mathcal{O}(T^{4/5})$ to $\mathcal{O}(T^{2/3})$ in their setting, and also provides a regret guarantee of $\mathcal{O}(\sqrt{T})$ in the restricted setting.[1]

## 2 Background

We now give the formal definition of the learning problem. We consider a sequential interaction scheme between a learner and an environment where in each round $t \in [T] = \{1, 2, \ldots, T\}$, the learner has to choose an action $V_t$ from a subset $\mathcal{S}_t$ of a known decision set $\mathcal{S} \subseteq \{0, 1\}^d$ with $\|v\|_1 \leq m$ for all $v \in \mathcal{S}$. We assume that the environment selects $\mathcal{S}_t$ according to some fixed (but unknown) distribution $\mathcal{P}$, independently of the interaction history. Unaware of the learner's decision, the environment also decides on a loss vector $\ell_t \in [0, 1]^d$ that will determine the loss suffered by the learner, which is of the form $V_t^\top \ell_t$. We make no assumptions on how the environment generates the sequence of loss vectors, that is, we are interested in algorithms that work in non-oblivious (or adaptive) environments. At the end of each round, the learner receives some feedback based on the loss vector and the action of the learner. The goal of the learner is pick its actions so as to minimize the losses it accumulates by the end of the $T$'th round. This setup generalizes the setting of online combinatorial optimization considered by Cesa-Bianchi and Lugosi [5], Audibert et al. [1], where the decision set is assumed to be fixed throughout the learning procedure. The interaction protocol is summarized on Figure 1 for reference.

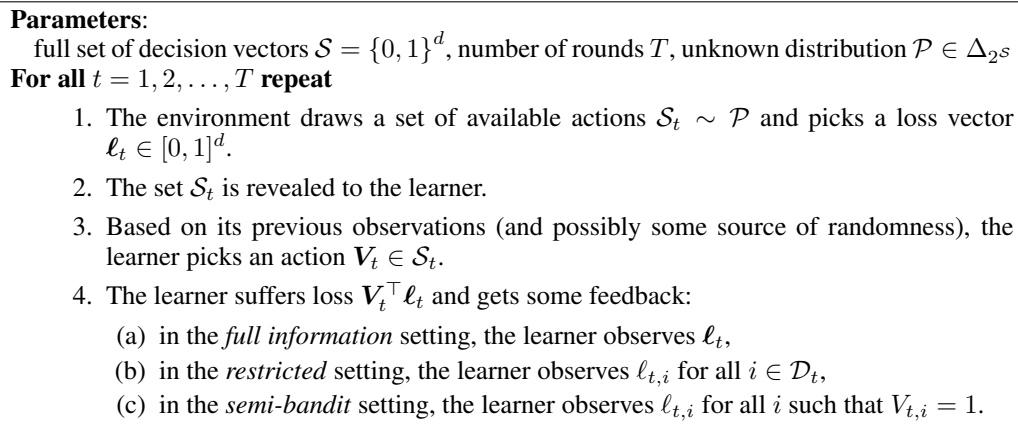

**Parameters**:
    full set of decision vectors $\mathcal{S} = \{0,1\}^d$, number of rounds $T$, unknown distribution $\mathcal{P} \in \Delta_{2^{\mathcal{S}}}$
**For all** $t = 1, 2, \ldots, T$ **repeat**

    1. The environment draws a set of available actions $\mathcal{S}_t \sim \mathcal{P}$ and picks a loss vector $\boldsymbol{\ell}_t \in [0,1]^d$.

    2. The set $\mathcal{S}_t$ is revealed to the learner.

    3. Based on its previous observations (and possibly some source of randomness), the learner picks an action $\boldsymbol{V}_t \in \mathcal{S}_t$.

    4. The learner suffers loss $\boldsymbol{V}_t^\top \boldsymbol{\ell}_t$ and gets some feedback:

        (a) in the *full information* setting, the learner observes $\boldsymbol{\ell}_t$,

        (b) in the *restricted* setting, the learner observes $\ell_{t,i}$ for all $i \in \mathcal{D}_t$,

        (c) in the *semi-bandit* setting, the learner observes $\ell_{t,i}$ for all $i$ such that $V_{t,i} = 1$.

Figure 1: The protocol of online combinatorial optimization with stochastic action availability.

We distinguish between three different feedback schemes, the simplest one being the *full information* scheme where the loss vectors are completely revealed to the learner at the end of each round. In the *restricted-information* scheme, we make a much milder assumption that the learner is informed about the losses of the *available* actions. Precisely, we define the set of *available components* as

$$\mathcal{D}_t = \{i \in [d] : \exists \boldsymbol{v} \in \mathcal{S}_t : v_i = 1\}$$

and assume that the learner can observe the $i$-th component of the loss vector $\boldsymbol{\ell}_t$ if and only if $i \in \mathcal{D}_t$. This is a sensible assumption in a number of practical applications, e.g., in sequential routing problems where components are associated with links in a network. Finally, in the *semi-bandit* scheme, we assume that the learner only observes losses associated with the components of its own decision, that is, the feedback is $\ell_{t,i}$ for all $i$ such that $V_{t,i} = 1$. This is the case in in online advertising settings where components of the decision vectors represent customer-ad allocations. The observation history $\mathcal{F}_t$ is defined as the sigma-algebra generated by the actions chosen by the learner and the decision sets handed out by the environment by the end of round $t$: $\mathcal{F}_t = \sigma(\boldsymbol{V}_t, \mathcal{S}_t, \ldots, \boldsymbol{V}_1, \mathcal{S}_1)$.

The performance of the learner is measured with respect to the best fixed *policy* (otherwise known as a *choice function* in discrete choice theory [16]) of the form $\boldsymbol{\pi} : 2^{\mathcal{S}} \to \mathcal{S}$. In words, a policy $\boldsymbol{\pi}$ will pick action $\boldsymbol{\pi}(\bar{\mathcal{S}}) \in \bar{\mathcal{S}}$ whenever the environment selects action set $\bar{\mathcal{S}}$. The (total expected) *regret* of the learner is defined as

$$R_T = \max_{\boldsymbol{\pi}} \sum_{t=1}^T \mathbb{E}\left[ (\boldsymbol{V}_t - \boldsymbol{\pi}(\mathcal{S}_t))^\top \boldsymbol{\ell}_t \right]. \tag{1}$$

Note that the above expectation integrates over *both* the randomness injected by the learner *and* the stochastic process generating the decision sets. The attentive reader might notice that this regret criterion is very similar to that of Kanade et al. [13], who study the setting of prediction with expert advice (where $m = 1$) and measure regret against the best fixed *ranking* of experts. It is actually easy to show that the optimal policy in their setting belongs to the set of ranking policies, making our regret definition equivalent to theirs.

## 3 Loss estimation by Counting Asleep Times

In this section, we describe our method used for estimating unobserved losses that works *without having to explicitly learn the availability distribution* $\mathcal{P}$. To explain the concept on a high level, let us now consider our simpler partial-observability setting, the restricted-information setting. For the formal treatment of the problem, let us fix any component $i \in [d]$ and define $A_{t,i} = \mathbb{1}_{\{i \in \mathcal{D}_t\}}$ and $a_i = \mathbb{E}[A_{t,i} | \mathcal{F}_{t-1}]$. Had we known the observation probability $a_i$, we would be able to estimate the $i$'th component of the loss vector $\boldsymbol{\ell}_t$ by $\hat{\ell}_{t,i}^* = (\ell_{t,i} A_{t,i})/a_i$, as the quantity $\ell_{t,i} A_{t,i}$ is observable. It is easy to see that the estimate $\hat{\ell}_{t,i}^*$ is unbiased by definition – but, unfortunately, we do not know $a_i$, so we have no hope to compute it. A simple idea used by Kanade et al. [13] is to devote

the first $T_0$ rounds of interaction solely to the purpose of estimating $a_i$ by the sample mean $\hat{a}_i = (\sum_{t=1}^{T_0} A_{t,i})/T_0$. While this trick gets the job done, it is obviously wasteful as we have to throw away all loss observations before the estimates are sufficiently concentrated. [2]

We take a much simpler approach based on the observation that the "asleep-time" of component $i$ is a geometrically distributed random variable with parameter $a_i$. The asleep-time of component $i$ starting from time $t$ is formally defined as

$$N_{t,i} = \min\{n > 0 : i \in \mathcal{D}_{t+n}\},$$

which is the number of rounds until the next observation of the loss associated with component $i$. Using the above definition, we construct our loss estimates as the vector $\hat{\boldsymbol{\ell}}_t$ whose $i$-th component is

$$\hat{\ell}_{t,i} = \ell_{t,i} A_{t,i} N_{t,i}. \tag{2}$$

It is easy to see that the above loss estimates are unbiased as

$$\mathbb{E}\left[\ell_{t,i} A_{t,i} N_{t,i} \,|\, \mathcal{F}_{t-1}\right] = \ell_{t,i}\mathbb{E}\left[A_{t,i} \,|\, \mathcal{F}_{t-1}\right]\mathbb{E}\left[N_{t,i} \,|\, \mathcal{F}_{t-1}\right] = \ell_{t,i} a_i \cdot \frac{1}{a_i} = \ell_{t,i}$$

for any $i$. We will refer to this loss-estimation method as *Counting Asleep Times* (CAT).

Looking at the definition (2), the attentive reader might worry that the vector $\hat{\boldsymbol{\ell}}_t$ *depends on future realizations of the random decision sets* and thus could be useless for practical use. However, observe that there is no reason that the learner should use the estimate $\hat{\ell}_{t,i}$ before component $i$ wakes up in round $t + N_{t,i}$ – which is precisely the time when the estimate becomes well-defined. This suggests a very simple implementation of CAT: whenever a component is not available, estimate its loss by the last observation from that component! More formally, set

$$\hat{\ell}_{t,i} = \begin{cases} \ell_{t,i}, & \text{if } i \in \mathcal{D}_t \\ \hat{\ell}_{t-1,i}, & \text{otherwise.} \end{cases}$$

It is easy to see that at the beginning of any round $t$, the two alternative definitions match for all components $i \in \mathcal{D}_t$. In the next section, we confirm that this property is sufficient for running our algorithm.

## 4    Algorithms & their analyses

For all information settings, we base our learning algorithms on the Follow-the-Perturbed-Leader (FPL) prediction method of Hannan [9], as popularized by Kalai and Vempala [12]. This algorithm works by additively perturbing the total estimated loss of each component, and then running an optimization oracle over the perturbed losses to choose the next action. More precisely, our algorithms maintain the cumulative sum of their loss estimates $\widehat{\boldsymbol{L}}_t = \sum_{s=1}^{t} \hat{\boldsymbol{\ell}}_t$ and pick the action

$$\boldsymbol{V}_t = \operatorname*{arg\,min}_{\boldsymbol{v} \in \mathcal{S}_t} \boldsymbol{v}^{\top}\left(\eta\widehat{\boldsymbol{L}}_{t-1} - \boldsymbol{Z}_t\right),$$

where $\boldsymbol{Z}_t$ is a perturbation vector with independent exponentially distributed components with unit expectation, generated independently of the history, and $\eta > 0$ is a parameter of the algorithm. Our algorithms for the different information settings will be instances of FPL that employ different loss estimates suitable for the respective settings. In the first part of this section, we present the main tools of analysis that will be used for each resulting method.

As usual for analyzing FPL-based methods [12, 10, 18], we start by defining a hypothetical forecaster that uses a time-independent perturbation vector $\widetilde{\boldsymbol{Z}}$ with standard exponential components and peeks one step into the future. However, we need an extra trick to deal with the randomness of the decision set: we introduce the time-independent decision set $\widetilde{\mathcal{S}} \sim \mathcal{P}$ (drawn independently of the filtration $(\mathcal{F}_t)_t$) and define

$$\widetilde{\boldsymbol{V}}_t = \operatorname*{arg\,min}_{\boldsymbol{v} \in \widetilde{\mathcal{S}}} \boldsymbol{v}^{\top}\left(\eta\widehat{\boldsymbol{L}}_t - \widetilde{\boldsymbol{Z}}\right).$$

Clearly, this forecaster is infeasible as it uses observations from the future. Also observe that $\widetilde{V}_{t-1} \sim V_t$ given $\mathcal{F}_{t-1}$. The following two lemmas show how analyzing this forecaster can help in establishing the performance of our actual algorithms.

**Lemma 1.** *For any sequence of loss estimates, the expected regret of the hypothetical forecaster against any fixed policy $\pi : 2^{\mathcal{S}} \to \mathcal{S}$ satisfies*

$$\mathbb{E}\left[\sum_{t=1}^{T} \left(\widetilde{V}_t - \pi(\widetilde{\mathcal{S}})\right)^{\mathsf{T}} \hat{\ell}_t\right] \leq \frac{m\left(\log d + 1\right)}{\eta}.$$

The statement is easily proved by applying the follow-the-leader/be-the-leader lemma[3] (see, e.g., [4, Lemma 3.1]) and using the upper bound $\mathbb{E}\left[\left\|\widetilde{Z}\right\|_{\infty}\right] \leq \log d + 1$.

The following result can be extracted from the proof of Theorem 1 of Neu and Bartók [18].

**Lemma 2.** *For any sequence of nonnegative loss estimates,*

$$\mathbb{E}\left[ (\widetilde{V}_{t-1} - \widetilde{V}_t)^{\mathsf{T}} \hat{\ell}_t \,\middle|\, \mathcal{F}_{t-1}\right] \leq \eta\, \mathbb{E}\left[\left(\widetilde{V}_{t-1}^{\mathsf{T}} \hat{\ell}_t\right)^2 \,\middle|\, \mathcal{F}_{t-1}\right].$$

In the next subsections, we apply these results to obtain bounds for the three information settings.

### 4.1 Algorithm for full information

In the simplest setting, we can use $\hat{\ell}_t = \ell_t$, which yields the following theorem:

**Theorem 1.** *Define*

$$L_T^* = \max\left\{\min_{\pi} \mathbb{E}\left[\sum_{t=1}^{T} \pi(\mathcal{S}_t)^{\mathsf{T}} \ell_t\right], 4(\log d + 1)\right\}.$$

*Setting $\eta = \sqrt{(\log d + 1)/L_T^*}$, the regret of FPL in the full information scheme satisfies*

$$R_T \leq 2m\sqrt{2 L_T^* (\log d + 1)}.$$

As this result is comparable to the best available bounds for FPL [10, 18] in the full information setting and a *fixed* decision set, it reinforces the observation of Kanade et al. [13], who show that the sleeping experts problem with full information and stochastic availability is no more difficult than the standard experts problem. The proof of Theorem 1 follows directly from combining Lemmas 1 and 2 with some standard tricks. For completeness, details are provided in Appendix A.

### 4.2 Algorithm for restricted feedback

In this section, we use the CAT loss estimate defined in Equation (2) as $\hat{\ell}_t$ in FPL, and call the resulting method SLEEPINGCAT. The following theorem gives the performance guarantee for this algorithm.

**Theorem 2.** *Define $Q_t = \sum_{i=1}^{d} \mathbb{E}\left[V_{t,i} \,|\, i \in \mathcal{D}_t\right]$. The total expected regret of SLEEPINGCAT against the best fixed policy is upper bounded as*

$$R_T \leq \frac{m(\log d + 1)}{\eta} + 2\eta m \sum_{t=1}^{T} Q_t.$$

*Proof.* We start by observing $\mathbb{E}\left[\pi(\widetilde{\mathcal{S}})^{\mathsf{T}} \hat{\ell}_t\right] = \mathbb{E}\left[\pi(\mathcal{S}_t)^{\mathsf{T}} \ell_t\right]$, where we used that $\hat{\ell}_t$ is independent of $\widetilde{\mathcal{S}}$ and is an unbiased estimate of $\ell_t$, and also that $\mathcal{S}_t \sim \widetilde{\mathcal{S}}$. The proof is completed by combining this with Lemmas 1 and 2, and the bound

$$\mathbb{E}\left[\left(\widetilde{V}_{t-1}^{\mathsf{T}} \hat{\ell}_t\right)^2 \,\middle|\, \mathcal{F}_{t-1}\right] \leq 2m Q_t.$$

The proof of this last statement follows from a tedious calculation that we defer to Appendix B. $\square$

Below, we provide two ways of further bounding the regret under various assumptions. The first one provides a universal upper bound that holds without any further assumptions.

**Corollary 1.** *Setting $\eta = \sqrt{(\log d + 1)/(2dT)}$, the regret of* SLEEPINGCAT *against the best fixed policy is bounded as*

$$R_T \le 2m\sqrt{2dT(\log d + 1)}.$$

The proof follows from the fact that $Q_t \le d$ no matter what $\mathcal{P}$ is. A somewhat surprising feature of our bound is its scaling with $\sqrt{d \log d}$, which is much worse than the logarithmic dependence exhibited in the full information case. It is easy to see, however, that this bound is not improvable in general – see Appendix D for a simple example. The next bound, however, shows that it is possible to improve this bound by assuming that most components are reliable in some sense, which is the case in many practical settings.

**Corollary 2.** *Assuming $a_i \ge \beta$ for all $i$, we have $Q_t \le 1/\beta$, and setting $\eta = \sqrt{\beta(\log d + 1)/(2T)}$ guarantees that the regret of* SLEEPINGCAT *against the best fixed policy is bounded as*

$$R_T \le 2m\sqrt{\frac{2T(\log d + 1)}{\beta}}.$$

### 4.3 Algorithm for semi-bandit feedback

We now turn our attention to the problem of learning with semi-bandit feedback where the learner only gets to observe the losses associated with its own decision. Specifically, we assume that the learner observes all components $i$ of the loss vector such that $V_{t,i} = 1$. The extra difficulty in this setting is that our actions influence the feedback that we receive, so we have to be more careful when defining our loss estimates. Ideally, we would like to work with unbiased estimates of the form

$$\hat{\ell}_{t,i}^* = \frac{\ell_{t,i}}{q_{t,i}^*} V_{t,i}, \qquad \text{where} \qquad q_{t,i}^* = \mathbb{E}\left[V_{t,i} | \mathcal{F}_{t-1}\right] = \sum_{\bar{S} \in 2^S} \mathcal{P}(\bar{S})\mathbb{E}\left[V_{t,i} \big| \mathcal{F}_{t-1}, \mathcal{S}_t = \bar{S}\right]. \quad (3)$$

for all $i \in [d]$. Unfortunately though, we are in no position to compute these estimates, as this would require perfect knowledge of the availability distribution $\mathcal{P}$! Thus we have to look for another way to compute reliable loss estimates. A possible idea is to use

$$q_{t,i} \cdot a_i = \mathbb{E}\left[V_{t,i} | \mathcal{F}_{t-1}, \mathcal{S}_t\right] \cdot \mathbb{P}\left[i \in \mathcal{D}_t\right].$$

instead of $q_{t,i}^*$ in Equation 3 to normalize the observed losses. This choice yields another unbiased loss estimate as

$$\mathbb{E}\left[\frac{\ell_{t,i}V_{t,i}}{q_{t,i}a_i} \bigg| \mathcal{F}_{t-1}\right] = \frac{\ell_{t,i}}{a_i}\mathbb{E}\left[\mathbb{E}\left[\frac{V_{t,i}}{q_{t,i}} \bigg| \mathcal{F}_{t-1}, \mathcal{S}_t\right]\bigg| \mathcal{F}_{t-1}\right] = \frac{\ell_{t,i}}{a_i}\mathbb{E}\left[A_{t,i} | \mathcal{F}_{t-1}\right] = \ell_{t,i}, \quad (4)$$

which leaves us with the problem of computing $q_{t,i}$ and $a_i$. While this also seems to be a tough challenge, we now show to estimate this quantity by generalizing the CAT technique presented in Section 3.

Besides our trick used for estimating the $1/a_i$'s by the random variables $N_{t,i}$, we now also have to face the problem of not being able to find a closed-form expression for the $q_{t,i}$'s. Hence, we follow the geometric resampling approach of Neu and Bartók [18] and draw an additional sequence of $M$ perturbation vectors $\boldsymbol{Z}_t'(1), \dots, \boldsymbol{Z}_t'(M)$ and use them to compute

$$\boldsymbol{V}_t'(k) = \arg\min_{\boldsymbol{v} \in \mathcal{S}_t}\left\{\eta\widehat{\boldsymbol{L}}_{t-1} - \boldsymbol{Z}_t'(k)\right\}$$

for all $k \in [M]$. Using these simulated actions, we define

$$K_{t,i} = \min\left(\left\{k \in [M] : V_{t,i}'(k) = V_{t,i}\right\} \cup \{M\}\right).$$

and

$$\hat{\ell}_{t,i} = \ell_{t,i}K_{t,i}N_{t,i}V_{t,i} \quad (5)$$

for all $i$. Setting $M = \infty$ makes this expression equivalent to $\frac{\ell_{t,i}V_{t,i}}{q_{t,i}a_i}$ in expectation, yielding yet another unbiased estimator for the losses. Our analysis, however, crucially relies on setting $M$ to

a finite value so as to control the variance of the loss estimates. We are not aware of any other work that achieves a similar variance-reduction effect without explicitly exploring the action space [17, 6, 5, 3], making this alternative bias-variance tradeoff a unique feature of our analysis. We call the algorithm resulting from using the loss estimates above SLEEPINGCATBANDIT. The following theorem gives the performance guarantee for this algorithm.

**Theorem 3.** *Define $Q_t = \sum_{i=1}^{d} \mathbb{E}\left[V_{t,i}\middle| i \in \mathcal{D}_t\right]$. The total expected regret of* SLEEPINGCATBANDIT *against the best fixed policy is bounded as*

$$R_T \leq \frac{m(\log d + 1)}{\eta} + 2\eta M m \sum_{t=1}^{T} Q_t + \frac{dT}{eM}.$$

*Proof.* First, observe that $\mathbb{E}\left[\hat{\ell}_{t,i}\middle|\mathcal{F}_{t-1}\right] \leq \ell_{t,i}$ as $\mathbb{E}\left[K_{t,i}V_{t,i}\middle|\mathcal{F}_{t-1},\mathcal{S}_t\right] \leq A_{t,i}$ and $\mathbb{E}\left[A_{t,i}N_{t,i}\middle|\mathcal{F}_{t-1}\right] = 1$ by definition. Thus, we can get $\mathbb{E}\left[\pi(\widetilde{\mathcal{S}})^\mathsf{T}\hat{\ell}_t\right] \leq \mathbb{E}\left[\pi(\mathcal{S}_t)^\mathsf{T}\ell_t\right]$ by a similar argument that we used in the proof of Theorem 2. After yet another long and tedious calculation (see Appendix C), we can prove

$$\mathbb{E}\left[\left(\widetilde{V}_{t-1}^\mathsf{T}\hat{\ell}_t\right)^2\middle|\mathcal{F}_{t-1}\right] \leq 2MmQ_t. \tag{6}$$

The proof is concluded by combining this bound with Lemmas 1 and 2 and the upper bound

$$\mathbb{E}\left[V_t^\mathsf{T}\ell_t\middle|\mathcal{F}_{t-1}\right] \leq \mathbb{E}\left[\widetilde{V}_{t-1}^\mathsf{T}\hat{\ell}_t\middle|\mathcal{F}_{t-1}\right] + \frac{d}{eM},$$

which can be proved by following the proof of Theorem 1 in Neu and Bartók [18]. □

**Corollary 3.** *Setting $\eta = \left(\frac{\sqrt{m}(\log d + 1)}{2dT}\right)^{2/3}$ and $M = \frac{1}{\sqrt{e}} \cdot \left(\frac{dT}{\sqrt{2}m(\log d + 1)}\right)^{1/3}$ guarantees that the regret of* SLEEPINGCATBANDIT *against the best fixed policy is bounded as*

$$R_T \leq (2mdT)^{2/3} \cdot (\log d + 1)^{1/3}.$$

The proof of the corollary follows from bounding $Q_t \leq d$ and plugging the parameters into the bound of Theorem 3. Similarly to the improvement of Corollary 2, it is possible to replace the factor $d^{2/3}$ by $(d/\beta)^{1/3}$ if we assume that $a_i \geq \beta$ for all $i$ and some $\beta > 0$.

This corollary implies that SLEEPINGCATBANDIT achieves a regret of $(2KT)^{2/3} \cdot (\log K + 1)^{1/3}$ in the case when $\mathcal{S} = [K]$, that is, in the $K$-armed sleeping bandit problem considered by Kanade et al. [13]. This improves their bound of $\mathcal{O}((KT)^{4/5} \log T)$ by a large margin, thanks to the fact that we did not have to explicitly learn the distribution $\mathcal{P}$.

## 5   Experiments

In this section we present the empirical evaluation of our algorithms for bandit and semi-bandit settings, and compare them to its counterparts [13]. We demonstrate that the wasteful exploration of BSFPL does not only result in worse regret bounds but also *degrades its empirical performance*.

For the bandit case, we evaluate SLEEPINGCATBANDIT using the same setting as Kanade et al. [13]. We consider an experiment with $T = 10,000$ and 5 arms, each of which are available independently of each other with probability $p$. Losses for each arm are constructed as random walks with Gaussian increments of standard deviation 0.002, initialized uniformly on $[0, 1]$. Losses outside $[0, 1]$ are truncated. In our first experiment (Figure 2, left), we study the effect of changing $p$ on the performance of BSFPL and SLEEPINGCATBANDIT. Notice that when $p$ is very low, there are few or no arms to choose from. In this case the problems are easy by design and all algorithms suffer low regret. As $p$ increases, the policy space starts to blow up and the problem becomes more difficult. When $p$ approaches one, it collapses into the set of single arms and the problem gets easier again. Observe that the behavior of SLEEPINGCATBANDIT follows this trend. On the other hand, the performance of BSFPL steadily decreases with increasing availability. This is due to the explicit exploration rounds in the main phase of BSFPL, that suffers the loss of the uniform policy scaled by the exploration probability. The performance of the uniform policy is plotted for reference.

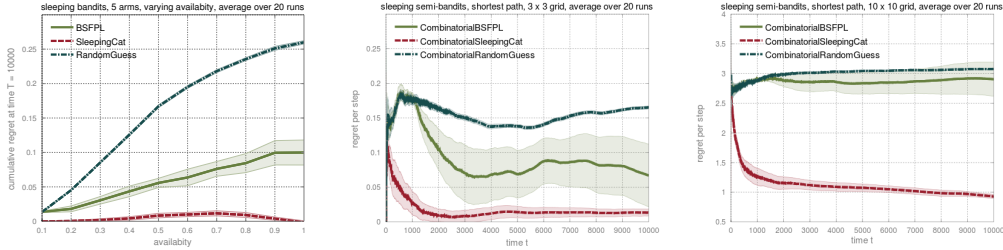

Figure 2: **Left:** Multi-arm bandits - varying availabilities. **Middle:** Shortest paths on a $3 \times 3$ grid. **Right:** Shortest paths on a $10 \times 10$ grid.

To evaluate SLEEPINGCATBANDIT in the semi-bandit setting, we consider the *shortest path problem* on grids of $3 \times 3$ and $10 \times 10$ nodes, which amounts to 12 and 180 edges respectively. For each edge, we generate a random-walk loss sequence in the same way as in our first experiment. In each round $t$, the learner has to choose a path from the lower left corner to the upper right one composed from available edges. The individual availability of each edge is sampled with probability 0.9, independently of the others. Whenever an edge gets disconnected from the source, it becomes unavailable itself, resuling in a quite complicated action-availability distribution. Once a learner chooses a path, the losses of chosen road segments are revealed and the learner suffers their sum.

Since [13] does not provide a combinatorial version of their approach, we compare against COMBB-SFPL, a straightforward extension of BSFPL. As in BSFPL, we dedicate an initial phase to estimate the availabilities of each component, requiring $d$ oracle calls per step. In the main phase, we follow BSFPL and alternate between exploration and exploitation. In exploration rounds, we test for the reachability of a randomly sampled edge and update the reward estimates as in BSFPL.

Figure 2 (middle and right) shows the performance of COMBBSFPL and SLEEPINGCATBANDIT for a fixed loss sequence, averaged over 20 samples of the component availabilities. We also plot the performance of a random policy that follows the perturbed leader with all-zero loss estimates. First observe that the initial exploration phase sets back the performance of COMBBSFPL significantly. The second drawback of COMBBSFPL is the explicit separation of exploration and the exploitation rounds. This drawback is far more apparent when the number of components increases, as it is the case for the $10 \times 10$ grid graph with 180 components. As COMBBSFPL only estimates the loss of one edge per exploration step, sampling each edge as few as 50 times eats up $9,000$ rounds from the available $10,000$. SLEEPINGCATBANDIT does not suffer from this problem as it uses all its observations in constructing the loss estimates.

## 6  Conclusions & future work

In this paper, we studied the problem of online combinatorial optimization with changing decision sets. Our main contribution is a novel loss-estimation technique that enabled us to prove strong regret bounds under various partial-feedback schemes. In particular, our results largely improve on the best known results for the sleeping bandit problem [13], which suffers large losses from both from an initial exploration phase and from explicit exploration rounds in the main phase. These findings are also supported by our experiments.

Still, one might ask if it is possible to efficiently achieve a regret of order $\sqrt{T}$ under semi-bandit feedback. While the EXP4 algorithm of Auer et al. [2] can be used to obtain such regret guarantee, running this algorithm is out of question as its time and space complexity can be double-exponential in $d$ (see also the comments in [15]). Had we had access to the loss estimates (3), we would be able to control the regret of FPL as the term on the right hand side of Equation (6) could be replaced by $md$, which is sufficient for obtaining a regret bound of $\mathcal{O}(m\sqrt{dT \log d})$. In fact, it seems that learning in the bandit setting requires significantly more knowledge about $\mathcal{P}$ than the knowledge of the $a_i$'s. The question if we can extend the CAT technique to estimate all the relevant quantities of $\mathcal{P}$ is an interesting problem left for future investigation.

**Acknowledgements**  The research presented in this paper was supported by French Ministry of Higher Education and Research, by European Community's Seventh Framework Programme (FP7/2007-2013) under grant agreement n°270327 (CompLACS), and by FUI project Hermès.

## Footnotes

[1]While not explicitly proved by Kanade et al. [13], their technique can be extended to work in the restricted setting, where it can be shown to guarantee a regret of $\mathcal{O}(T^{3/4})$.

[2]Notice that we require "sufficient concentration" from $1/\hat{a}_i$ and not only from $\hat{a}_i$! The deviation of such quantities is rather difficult to control, as demonstrated by the complicated analysis of Kanade et al. [13].

[3]This lemma can be proved in the current case by virtue of the fixed decision set $\widetilde{\mathcal{S}}$, allowing the necessary recursion steps to go through.

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
