[Supplementary Material]

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

# A Proof of Theorem 1

We begin by applying Lemma 1, exploiting that $\hat{\ell}_t = \ell_t$ and $\widetilde{\mathcal{S}}$ is an independent copy of $\mathcal{S}_t$:

$$\sum_{t=1}^{T} \mathbb{E}\left[\widetilde{V}_t^{\mathsf{T}}\ell_t\right] - \sum_{t=1}^{T} \mathbb{E}\left[\pi(\mathcal{S}_t)^{\mathsf{T}}\ell_t\right] = \mathbb{E}\left[\sum_{t=1}^{T}\left(\widetilde{V}_t - \pi(\widetilde{\mathcal{S}})\right)^{\mathsf{T}}\hat{\ell}_t\right] \le \frac{m\left(\log d + 1\right)}{\eta}$$

Next, we apply Lemma 2 to obtain

$$\mathbb{E}\left[(V_t - \widetilde{V}_t)^{\mathsf{T}}\hat{\ell}_t\right] \le \eta \, \mathbb{E}\left[\left(\widetilde{V}_{t-1}^{\mathsf{T}}\hat{\ell}_t\right)^2\right] \le \eta m \mathbb{E}\left[V_t^{\mathsf{T}}\ell_t\right],$$

where we used that $\widetilde{V}_{t-1} \sim V_t$ and $\widetilde{V}_{t-1}^{\mathsf{T}}\ell_t \le m$. Introducing the notation

$$C_T = \sum_{t=1}^{T} \mathbb{E}\left[V_t^{\mathsf{T}}\ell_t\right],$$

we get by combining the above bounds that

$$C_T - L_T^* \le \frac{m\left(\log d + 1\right)}{\eta} + \eta m C_T.$$

After reordering, we get

$$C_T - L_T^* \le \frac{1}{1 - m\eta}\left(\frac{m\left(\log d + 1\right)}{\eta} + \eta m L_T^*\right).$$

The bound follows from plugging in the choice of $\eta$ and observing that $1 - m\eta \ge 1/2$ holds by the assumption of the theorem.

# B Proof details for Theorem 2

In the restricted information case, the term on the right hand side of the bound of Lemma 2 can be upperbounded as follows:

$$\mathbb{E}\left[\left(\widetilde{V}_{t-1}^{\mathsf{T}}\hat{\ell}_t\right)^2 \middle| \mathcal{F}_{t-1}\right] = \mathbb{E}\left[\sum_{j=1}^{d}\sum_{k=1}^{d}\left(\widetilde{V}_{t-1,j}\hat{\ell}_{t,j}\right)\left(\widetilde{V}_{t-1,k}\hat{\ell}_{t,k}\right)\middle| \mathcal{F}_{t-1}\right]$$

$$\le \mathbb{E}\left[\sum_{j=1}^{d}\sum_{k=1}^{d}\frac{N_{t,j}^2 + N_{t,k}^2}{2}\left(\widetilde{V}_{t-1,j}A_{t,j}\ell_{t,j}\right)\left(\widetilde{V}_{t-1,k}A_{t,k}\ell_{t,k}\right)\middle| \mathcal{F}_{t-1}\right]$$

(using the definition of $\hat{\ell}_t$ and $2AB \le A^2 + B^2$)

$$= \mathbb{E}\left[\sum_{j=1}^{d}\sum_{k=1}^{d}N_{t,j}^2\left(\widetilde{V}_{t-1,j}A_{t,j}\ell_{t,j}\right)\left(\widetilde{V}_{t-1,k}A_{t,k}\ell_{t,k}\right)\middle| \mathcal{F}_{t-1}\right]$$

(by symmetry)

$$\le 2\mathbb{E}\left[\sum_{j=1}^{d}\frac{1}{a_j^2}\left(\widetilde{V}_{t-1,j}A_{t,j}\ell_{t,j}\right)\sum_{k=1}^{d}\widetilde{V}_{t-1,k}\ell_{t,k}\middle| \mathcal{F}_{t-1}\right]$$

(using $\mathbb{E}\left[N_{t,j}^2 \middle| \mathcal{F}_{t-1}\right] = (2 - a_j)/a_j^2 \le 2/a_j^2$)

$$= 2m\mathbb{E}\left[\sum_{j=1}^{d}\frac{1}{a_j}\left(\widetilde{V}_{t-1,j}\ell_{t,j}\right)\middle| \mathcal{F}_{t-1}\right]$$

(using $\left\|\widetilde{V}_t\right\|_1 \le m$ and $\mathbb{E}\left[A_{t,j}\middle| \mathcal{F}_{t-1}\right] = a_j$

$$\le 2m\sum_{j=1}^{d}\mathbb{E}\left[V_{t,j}\middle| j \in \mathcal{D}_t, \mathcal{F}_{t-1}\right],$$

where in the last line, we used that $\widetilde{V}_{t-1}$ is identically distributed as $V_t$.

## C   Proof details for Theorem 3

In the semi-bandit case, the term on the right hand side of the bound of Lemma 2 can be upper-bounded as follows:

$$\mathbb{E}\left[\left(\widetilde{\boldsymbol{V}}_{t-1}^{\mathsf{T}}\hat{\boldsymbol{\ell}}_t\right)^2\middle|\mathcal{F}_{t-1}\right] = \mathbb{E}\left[\sum_{j=1}^{d}\sum_{k=1}^{d}\left(\widetilde{V}_{t-1,j}\hat{\ell}_{t,j}\right)\left(\widetilde{V}_{t-1,k}\hat{\ell}_{t,k}\right)\middle|\mathcal{F}_{t-1}\right]$$

$$\leq \mathbb{E}\left[\sum_{j=1}^{d}\sum_{k=1}^{d}\frac{K_{t,j}^2+K_{t,k}^2}{2}\cdot\frac{N_{t,j}^2+N_{t,k}^2}{2}\left(\widetilde{V}_{t-1,j}V_{t,j}\ell_{t,j}\right)\left(\widetilde{V}_{t-1,k}V_{t,k}\ell_{t,k}\right)\middle|\mathcal{F}_{t-1}\right]$$

(using the definition of $\hat{\boldsymbol{\ell}}_t$ and $2AB \leq A^2 + B^2$)

$$= \mathbb{E}\left[\sum_{j=1}^{d}\sum_{k=1}^{d}\frac{K_{t,j}^2N_{t,j}^2+K_{t,j}^2N_{t,k}^2+K_{t,k}^2N_{t,j}^2+K_{t,k}^2N_{t,k}^2}{4}\left(\widetilde{V}_{t-1,j}V_{t,j}\ell_{t,j}\right)\left(\widetilde{V}_{t-1,k}V_{t,k}\ell_{t,k}\right)\middle|\mathcal{F}_{t-1}\right]$$

$$= \mathbb{E}\left[\sum_{j=1}^{d}\frac{K_{t,j}^2N_{t,j}^2}{2}\left(\widetilde{V}_{t-1,j}V_{t,j}\ell_{t,j}\right)\sum_{k=1}^{d}\left(\widetilde{V}_{t-1,k}V_{t,k}\ell_{t,k}\right)\middle|\mathcal{F}_{t-1}\right]$$

$$+ \frac{1}{2}\cdot\mathbb{E}\left[\sum_{j=1}^{d}K_{t,j}^2\left(\widetilde{V}_{t-1,j}V_{t,j}\ell_{t,j}\right)\sum_{k=1}^{d}N_{t,k}^2\left(\widetilde{V}_{t-1,k}V_{t,k}\ell_{t,k}\right)\middle|\mathcal{F}_{t-1}\right]$$

$$\leq \frac{m}{2}\cdot\mathbb{E}\left[\mathbb{E}\left[\sum_{j=1}^{d}MK_{t,j}N_{t,j}^2\left(\widetilde{V}_{t-1,j}V_{t,j}\ell_{t,j}\right)\middle|\mathcal{F}_{t-1},\mathcal{S}_t\right]\middle|\mathcal{F}_{t-1}\right]$$

$$+ \frac{1}{2}\cdot\mathbb{E}\left[\mathbb{E}\left[\sum_{j=1}^{d}MK_{t,j}\left(\widetilde{V}_{t-1,j}V_{t,j}\ell_{t,j}\right)\sum_{k=1}^{d}N_{t,k}^2\left(\widetilde{V}_{t-1,k}A_{t,k}\ell_{t,k}\right)\middle|\mathcal{F}_{t-1},\mathcal{S}_t\right]\middle|\mathcal{F}_{t-1}\right]$$

(using $K_{t,j} \leq M$ and $V_{t,k} \leq A_{t,k}$ and $\left\|\widetilde{\boldsymbol{V}}_t\right\|_1 \leq m$)

$$\leq \frac{m}{2}\cdot\mathbb{E}\left[\sum_{j=1}^{d}MN_{t,j}^2\left(\widetilde{V}_{t-1,j}A_{t,j}\ell_{t,j}\right)\middle|\mathcal{F}_{t-1}\right]$$

$$+ \frac{1}{2}\cdot\mathbb{E}\left[\sum_{j=1}^{d}M\left(\widetilde{V}_{t-1,j}A_{t,j}\ell_{t,j}\right)\sum_{k=1}^{d}N_{t,k}^2\left(\widetilde{V}_{t-1,k}A_{t,k}\ell_{t,k}\right)\middle|\mathcal{F}_{t-1}\right]$$

(using $\mathbb{E}\left[K_{t,j}V_{t,j}\middle|\mathcal{F}_{t-1},\mathcal{S}_t\right] \leq A_{t,j}$ by definition of $K_{t,j}$ and independence of $K_{t,j}$ and $V_{t,j}$)

$$\leq 2Mm\mathbb{E}\left[\sum_{j=1}^{d}\frac{1}{a_j^2}\left(\widetilde{V}_{t-1,j}A_{t,j}\ell_{t,j}\right)\middle|\mathcal{F}_{t-1}\right]$$

(using $\left\|\widehat{\boldsymbol{V}}_t\right\|_1 \leq m$ and $\mathbb{E}\left[N_{t,j}^2\middle|\mathcal{F}_{t-1}\right] = (2-a_j)/a_j^2 \leq 2/a_j^2$)

$$= 2Mm\mathbb{E}\left[\sum_{j=1}^{d}\frac{1}{a_j}\left(\widetilde{V}_{t-1,j}\ell_{t,j}\right)\middle|\mathcal{F}_{t-1}\right]$$

$$\leq 2Mm\sum_{j=1}^{d}\mathbb{E}\left[V_{t,j}\middle|j\in\mathcal{D}_t,\mathcal{F}_{t-1}\right],$$

where in the last line, we used that $\widetilde{\boldsymbol{V}}_{t-1}$ is identically distributed as $\boldsymbol{V}_t$.

## D  A lower bound for restricted feedback

Consider a sleeping experts problem with $d$ experts with loss sequence $(\ell_t)_t$. In each round $t = 1, 2, \ldots, T$ the learner picks $I_t$. For simplicity, assume that $d$ is even and let $N = d/2$. Let $\mathcal{P}$ be such that it assigns a probability of $1/N$ to each pair $(2i - 1, 2i)$ of experts, that is, only two experts are awake at each time. The regret of any learning algorithm in this problem can be written as

$$R_T = \max_\pi \mathbb{E}\left[\sum_{t=1}^T \left(\ell_{t,I_t} - \ell_{t,\pi(\mathcal{S}_t)}\right)\right] = \max_\pi \mathbb{E}\left[\sum_{j=1}^N \sum_{t=1}^T \mathbb{1}_{\{2i \in \mathcal{S}_t\}} \left(\ell_{t,I_t} - \ell_{t,\pi(\mathcal{S}_t)}\right)\right].$$

We now define $N$ full-information games $G_1, \ldots, G_N$ with two experts each as follows: In game $G_i$, the number of rounds is $T_i = \sum_{t=1}^T \mathbb{1}_{\{2i \in \mathcal{S}_t\}}$, the decision of the learner in round $t$ is $J_t$ and the sequence of loss functions is $(\ell_t(i)_t)$ so that the regret in game $G_i$ is defined as

$$R_T(i) = \max_j \sum_{t=1}^{T_i} \left(\ell_{t,J_t}(i) - \ell_{t,j}(i)\right).$$

It is well-known (e.g., [4, Section 3.7]) that there exists a distribution of losses that guarantees that $\mathbb{E}[R_T(i)] \geq c\sqrt{T_i}$ for some constant $c > 0$, no matter what algorithm the learner uses. The result follows from observing that there exists a mapping between the full-information games $G_1, \ldots, G_N$ and our original problem such that

$$\mathbb{E}\left[\left.\sum_{j=1}^N \sum_{t=1}^T \mathbb{1}_{\{2i \in \mathcal{S}_t\}} \left(\ell_{t,I_t} - \ell_{t,\pi(\mathcal{S}_t)}\right)\right| (\mathcal{S}_t)_{t=1}^T\right] = \sum_{i=1}^N \mathbb{E}[R_T(i)] \geq c \sum_{i=1}^N \sqrt{T_i}.$$

That, is we get that

$$R_T \geq c \sum_{i=1}^N \mathbb{E}\left[\sqrt{T_i}\right].$$

As $\lim_{T\to\infty} \frac{T}{T_i} = N$ holds almost surely, we get that

$$\lim_{T\to\infty} \frac{R_T}{\sqrt{TN}} \geq c,$$

showing that there exist sleeping experts problems for any even $d$ where the guarantees of Corollary 1 cannot be improved asymptotically.