[Reviews · NeurIPS 2014]

Submitted by Assigned_Reviewer_6

Problem definition:- The paper studies online combinatorial optimization with stochastic decision sets and adversarial losses. In this problem a subset S_t of available actions of a decision set S\subseteq {0,1}^d is chosen stochastically. Then the algorithm chooses an action v\in S_t and incurs a loss of v^T*l_t where l_t is a loss vector chosen by an oblivious adversary. The paper studies the problem in three settings full information, semi-bandit and a new setting which they term restricted.

Results: Previously results were known in the full information and semi-bandit setting with sublinear regret bounds. The paper contributes in two ways.
a) It introduces a way of getting an unbiased estimate of the loss vector. They call this "Counting awake times". This along with the follow the perturbed leader allows them to improves the regret bound for the semi-bandit setting previously known. The previous result achieved sublinear regret by separating exploration from exploitation.
b) It also shows new results for the restricted setting.

Quality of the paper:- The techniques appear reasonably sound.

Clarity of the paper:- The paper is reasonably well written.

Originality:- The most novel aspect of the paper is the new way of estimating unbiased estimate of the loss vector. It does look non-trivial and doesn't seem to be exist in the current literature.

Significance:- The paper improves the regret bound of a reasonably general problem. The paper does mention some applications at the beginning. The one assumption in the model which might make it a bit unusable for practical applications is that the decision sets are chosen i.i.d from a distribution. I don't see why this should happen for any realistic application.
Summary: The paper studies Online combinatorial optimization with stochastic
decision sets and adversarial losses. It introduces a way of getting unbiased estimates by a new technique called "Couting Awake Times".

Submitted by Assigned_Reviewer_25

The paper is clear and well written.
The paper proposes an algorithm for a problem close to the sleeping experts. As in the case of sleeping experts, the decision set is available stochastically and the rewards are chosen by a non oblivious adversary. Here, the goal of the learner is to sequentially select its actions from a combinatorial decision set.
The sleeping expert problem is important in practice, and the combinatorial optimization corresponds to the case where the decision set is structured by a list of actions, for instance in the case of advertising, the list of ads to display for a profile.
As in the case of sleeping experts, the proposed algorithm is based on the well-known Follow-the-Perturbed-Leader algorithm. It is analyzed for three different feedbacks: full information, restricted information (the value of the loss function is revealed for available actions), and semi-bandit (the feedback is revealed for chosen actions). As in the case of sleeping expert, the regret bounds are given with respect to the best fixed policy.
The originality of the paper lies in the way in which the availability of decisions is taken into account. The authors propose a smart and efficient method to obtain unbiased loss estimation when the availability distribution is unknown: when an action becomes unavailable, its loss is estimated by its last observation. The reason of this simple approach is well explained and analyzed. This loss estimation allows improving experimental results of the previous approach BSFPL (sleeping experts) since there is no exploration time to estimate the availability of actions.
However, except the availability of decisions, the algorithms and the mathematical framework come from “An Efficient Algorithm for Learning with Semi-Bandit Feedback”, which is given as reference. To sum up, this work is an adaptation of algorithms proposed in the above paper to the sleeping expert problem.
Furthermore, the paper is not self-sufficient, and to reproduce the analysis, the reader has to extract some statements in the demonstrations of the above paper, which are used here.
I suggest that the authors provide all the demonstrations in the extended version. To improve the paper, I also suggest that the distinction between the decision vector and the action is done: the learner chooses a decision vector $V_t$, where each dimension $V_{i,t}$ is an action.
There are some typos: line 211, and line 232.
Summary: The most significant shortcoming of this paper is the lack of originality, since most of this work has been published. However, this paper addresses an interesting problem. It is technically sound, clear and well written. I tend to vote for accepting it.

Submitted by Assigned_Reviewer_36

The paper considers an online learning problem with a changing decision
set and adversarial payoffs. More precisely the decision set is always a
subset of some space S, typically {0, 1}^d. At each time only a subset S_t
of the decision set is available from which the decision maker has to
make a choice. The loss vector is some vector $\ell$ and the actual loss
is \ell \cdot d, where d is the decision vector. The paper considers the
expert, semi-bandit and bandit setting. The subsets S_t that give the
feasible set at each round are assumed to be decided stochastically
according to some fixed distribution by the environment. The payoffs on
the other hand may be designed by an adversary.

The paper presents some nice algorithms that improve the state of the art
for the semi-bandit and bandit setting. The main idea is constructing
better unbiased estimators for the loss vectors. This builds on some
recent prior work, combined with follow the perturbed leader (FPL). I
think the trick is quite neat and may be useful in other problems. As a
consequence of the (generic) algorithm for the combinatorial (semi)-bandit
problem, the paper also makes progress on algorithms for sleeping experts
problem.
Summary: The paper considers combinatorial (semi)-bandits question with partial
stochastic availability and linear losses. Here the feasible set (which
changes at each round) is some subset of S^t, which is chosen from some
fixed but unknown distribution. The losses are allowed to be adversarial.
The authors provide a cool FPL-based algorithm that achieves sub-linear
regret.

Author Feedback
Author rebuttal: We thank the reviewers for their helpful comments. Below, we reply to each of the specific points made by the reviewers.

Reviewer 1
----------
1. Relations to Neu & Bartok (2013)
We would like to point out that the main challenge in our setting is defining appropriate loss estimates. These estimates have to take two sources of uncertainty into account: 1) availability and 2) the random choices of the learning algorithm itself. For treating the random choices of the learner, we do build on Neu & Bartok (2013) and we properly credit them. For the availability we use a novel estimation trick to circumvent the initial exploration phase of Kanade et al. (2009). This trick is core to our approach, and its analysis does not follow from any previous work.

Furthermore, note that (just like EXP3/Hedge) FPL is a widely studied learning algorithm that forms the basis of many advanced algorithms for online learning. We build on the analysis of Neu & Bartok (2013) for proving our bounds simply because their technique provides the strongest known bounds for FPL in the combinatorial setting.

We are ready to provide more proof details in the appendix if this helps in making our results easier to verify.

2. Improving experimental results
We would like to point out that our main result also implies a significant theoretical improvement over BSFPL.

3. Actions vs decisions
In fact, we inherited the use of the word "action" from the classical references on Combinatorial Bandits (Cesa-Bianchi and Lugosi, 2012, Audibert et al., 2014), where it refers to elements of the decision space S\subseteq R^d. We refer to individual components of actions simply as "components".

Reviewer 3
----------
1. IID decision sets
Some realistic situations where IID decision sets may arise are when actions correspond to measurements from physical sensors that might be corrupted by thermal noise, or might not be available thanks to IID channel noise. That said, we agree that the stochastic assumption on the decision sets is limiting for practical applications, but it seems that this assumption is essential for constructing computationally efficient algorithms. Indeed, recent results of Kanade & Steinke (2012) suggest that finding efficient algorithms for learning with adversarial decision sets and adversarial losses is in some sense very tough problem. As our main concern in this paper is computational efficiency, we accept the stochastic assumption on the decision sets, and show that these assumptions are sufficient for efficiency even when the decision space is combinatorial. It is interesting to note that while the running time of EXP4 is super-exponential in d for combinatorial decision sets, our algorithms can be implemented in polytime whenever there is a polytime solver for the offline optimization problem. We will clarify these issues in our final version.